# Inhibition of Adult Neurogenesis in Male Mice after Repeated Exposure to Paracetamol Overdose

**DOI:** 10.3390/ijms25041964

**Published:** 2024-02-06

**Authors:** Juan Suárez, Marialuisa de Ceglia, Miguel Rodríguez-Pozo, Antonio Vargas, Ignacio Santos, Sonia Melgar-Locatelli, Adriana Castro-Zavala, Estela Castilla-Ortega, Fernando Rodríguez de Fonseca, Juan Decara, Patricia Rivera

**Affiliations:** 1Departamento de Anatomía Humana, Medicina Legal e Historia de la Ciencia, Facultad de Medicina, Universidad de Málaga, 29071 Málaga, Spain; juan.suarez@uma.es (J.S.); rpmiguel@uma.es (M.R.-P.); isantos@uma.es (I.S.); 2Grupo de Neuropsicofarmacología, Instituto IBIMA-Plataforma BIONAND, Unidad de Gestión Clínica de Salud Mental, Hospital Regional Universitario de Málaga, Av. de Carlos Haya, 29010 Málaga, Spain; marialuisa.deceglia@ibima.eu (M.d.C.); antonio.vargas@ibima.eu (A.V.); 0619587398@uma.es (S.M.-L.); adriana.castro@uma.es (A.C.-Z.); estela.castilla@ibima.eu (E.C.-O.); fernando.rodriguez@ibima.eu (F.R.d.F.); 3Departamento de Psicobiología y Metodología de las Ciencias del Comportamiento, Facultad de Psicología, Universidad de Málaga, 29010 Málaga, Spain; 4Unidad Clínica de Neurología, Hospital Regional Universitario de Málaga, Instituto IBMA-Plataforma BIONAND, 29010 Málaga, Spain

**Keywords:** adult neurogenesis, acetaminophen, paracetamol, endocannabinoid, subgranular zone, hypothalamus, toxicology

## Abstract

Paracetamol, or acetaminophen (N-acetyl-para-aminophenol, APAP), is an analgesic and antipyretic drug that is commonly used worldwide, implicated in numerous intoxications due to overdose, and causes serious liver damage. APAP can cross the blood–brain barrier and affects brain function in numerous ways, including pain signals, temperature regulation, neuroimmune response, and emotional behavior; however, its effect on adult neurogenesis has not been thoroughly investigated. We analyze, in a mouse model of hepatotoxicity, the effect of APAP overdose (750 mg/kg/day) for 3 and 4 consecutive days and after the cessation of APAP administration for 6 and 15 days on cell proliferation and survival in two relevant neurogenic zones: the subgranular zone of the dentate gyrus and the hypothalamus. The involvement of liver damage (plasma transaminases), neuronal activity (c-Fos), and astroglia (glial fibrillar acidic protein, GFAP) were also evaluated. Our results indicated that repeated APAP overdoses are associated with the inhibition of adult neurogenesis in the context of elevated liver transaminase levels, neuronal hyperactivity, and astrogliosis. These effects were partially reversed after the cessation of APAP administration for 6 and 15 days. In conclusion, these results suggest that APAP overdose impairs adult neurogenesis in the hippocampus and hypothalamus, a fact that may contribute to the effects of APAP on brain function.

## 1. Introduction

Paracetamol, or acetaminophen (N-acetyl-para-aminophenol, APAP), is one of the most used drugs worldwide, famous for its analgesic and antipyretic actions. Available in a wide variety of pharmaceutical forms, APAP is principally administered for the treatment of cold, fever, and acute or chronic pain, especially in patients in whom non-steroidal anti-inflammatory drugs (NSAIDs) are contraindicated [1]. Nonetheless, recent evidence estimates that APAP causes 6% of poisonings worldwide [2] because of acute and/or chronic overdose. Acute overdose is a common form of suicide because of APAP’s low cost and accessibility; chronic overdose is usually unintentional and happens as a result of therapeutic misuse and prolonged excessive dosing [3]. The most dangerous outcome of APAP poisoning is severe liver injury caused by the metabolic conversion to N-acetyl-p-benzoquinone imine (NAPQI), which results in glutathione (GSH) depletion and covalent binding to proteins, subsequently leading to protein nitration and mitochondrial permeability transition and ultimately inducing oxidative stress, cell death, and loss of liver function [4]. APAP overdose also causes alterations in brain functions that have long been considered side effects of liver damage [5]. Thus, hepatic encephalopathy (HE), in turn, indicates a worsening of the liver’s functionality. The pathophysiology of HE is variable, including ammonia accumulation, oxidative stress, glutamine signaling, and inflammatory factors, highlighting the excess of NH_4_^+^ in the blood derived from hepatocellular damage, which spreads to the central nervous system (CNS), joining the glutamine synthesis route with a cytotoxic nature [3,6,7,8].

However, paracetamol not only affects the brain through its relationship with liver damage but also has a direct effect on the brain.

APAP is a moderately lipid-soluble weak organic acid that is transported in its non-ionized form in the blood, so its hydrophobic nature allows it to readily penetrate cellular membranes [9]. Since the effects of this drug must be exhibited by acting on the CNS, several studies have focused on verifying whether APAP is capable of diffusing through the blood–brain barrier (BBB). It has been shown that paracetamol can easily pass through it and be found in the cerebrospinal fluid of healthy individuals [10]. Furthermore, it has been shown that the drug is distributed homogeneously throughout the different areas of the brain and spinal cord and does not have a greater affinity or action in any specific region [11]. While its hepatic metabolism and how its overdose leads to hepatotoxicity are well established, its mechanism of action in the brain is still under debate [12].

APAP mainly acts in the CNS, reducing the active form of cyclooxygenase (COX) 1 and 2 and dampening prostaglandin release and the activation of the inflammatory response [3]. Unlike other NSAIDs, APAP does not inhibit COX activity in the periphery, which reduces its side effects. Recent evidence shows that APAP efficacy may depend on its transformation into N-arachidonoylphenolamine (AM404), an active metabolite that activates transient receptor potential vanilloid-1 (TRPV1) and inhibits endocannabinoid anandamide (AEA) reuptake from the synaptic cleft [13]. However, recent evidence demonstrates that brain alterations occurring after APAP overdose may depend on on-site APAP metabolism into toxic metabolites, GSH depletion, the generation of oxidative stress, and a dose-dependent increase in cell mortality [3].

Neurogenesis is a life-long process by which neural progenitors or stem cells proliferate and differentiate into new neurons. The subventricular zone (SVZ) and subgranular zone (SGZ) are the best-characterized neurogenic niches in the adult rodent brain; however, adult neurogenesis in rodents is not restricted to these areas, which can be demonstrated in other brain regions such as the hypothalamus, substantial nigra, and amygdala [14].

A large body of evidence suggests the role of APAP overdose in neurodevelopmental impairment [15,16,17]. Nevertheless, few studies relate adult neurogenesis to paracetamol overdose toxicity. In this sense, it is known that APAP metabolite AM404 is a suppressor of hippocampal cell proliferation [18] and can affect hippocampal neurogenesis through TRPV-1-dependent mechanisms [19].

This study aims to evaluate the effect of repeated overdoses of APAP on the proliferation of neural stem cells and the survival of mature neurons at different time points (6 h, 6 days, and 15 days) to evaluate its impact on adult neurogenesis in the SGZ of the hippocampus and the hypothalamus, brain regions related to cognitive processes (learning and memory) and metabolic functions, respectively [14]. Also, we evaluated APAP overdose effects on c-Fos expression in the areas of interest, knowing that its activation is required for synaptic plasticity and learning [20]. Moreover, considering previous evidence obtained in our laboratory that suggests that paracetamol-induced liver injury provokes alterations in inflammatory pathways [21,22], we studied the expression of glial fibrillar acidic protein (GFAP), which is a recognized marker of neuroinflammation, as well as an astrocyte precursor in adult neurogenic niches [23]. Overall, our results suggested that repeated overdoses of APAP are associated with the inhibition of adult neurogenesis in the context of high liver transaminase levels, neuronal hyperactivity, and astrogliosis. These effects were most evident in the hypothalamus and were partially reversed after the cessation of APAP administration for 6 and 15 days.

## 2. Results

### 2.1. Effect of APAP Overdose on Cell Proliferation in the Subgranular Zone of the Dentate Gyrus and Hypothalamus

To investigate the impact of repeated APAP administration (750 mg/kg) on cell proliferation in the relevant neurogenic zones of the adult brain, we evaluated the number of newborn cells in the SGZ of the dentate gyrus (DG) and hypothalamus after the administration of 5′-bromo-2′-deoxyuridine (BrdU) (50 mg/kg) for three consecutive days before sacrifice. The number of BrdU-immunoreactive (BrdU-ir) cells was differentially detected depending on the experimental group (the APAP injections and time of sacrifice after the last APAP administration) and the neurogenic zone analyzed.

One-way ANOVA showed the effect of repeated APAP administration in both neurogenic areas analyzed, the SGZ (*p* < 0.05) and hypothalamus (*p* < 0.001) (Figure 1). The number of BrdU-ir cells was lower in the hypothalamus, but not in the SGZ, after repeated APAP administration for three consecutive days (APAPx3) compared with the control group (** *p* < 0.01; Figure 1B,I).

Likewise, the number of BrdU-ir cells was lower in the SGZ and hypothalamus when animals received APAP for four consecutive days and were sacrificed 6 h after the last administration (APAPx4–6 h) compared with the control group (*** *p* < 0.05/0.001; Figure 1A,B,E,J).

Interestingly, a significant increase in the number of BrdU-ir cells was detected in the hypothalamus of mice that received APAP for 4 consecutive days and were sacrificed 15 days after the last administration (APAPx4–15 days) compared with the APAPx4–6 h group (^$$^ *p* < 0.01; Figure 1B,L).

### 2.2. Effect of APAP Overdose on Cell Survival in the Hippocampus and Hypothalamus

To investigate the impact of repeated APAP administration on cell survival in adult mouse brains, we evaluated the number of surviving cells after 5′-iodo-2′-deoxyuridine (IdU) administration (42.75 mg/kg) for three consecutive days starting on the second day of APAP administration. The IdU-ir cells in the hippocampus and hypothalamus of mice sacrificed 6 and 15 days after administering APAPx4 were analyzed (Figure 2).

One-way ANOVA showed the effect of APAP administration on cell survival in the hypothalamus (*p* < 0.01) but not in the dentate gyrus (DG) (Figure 2). The DG of mice repeatedly treated with APAP did not show any difference in the number of IdU-ir cells compared with that of the control mice (Figure 2A,C–E). However, we observed a significant decrease in the number of IdU-ir cells specifically in the hypothalamus of APAPx4-treated mice sacrificed 6 days after the last APAP administration compared with the control group (** *p* < 0.01; Figure 2B,G).

### 2.3. Effect of APAP Overdose on Neuronal Activity in the Hippocampus and Hypothalamus

To investigate the impact of repeated APAP administration on neuronal activity, we evaluated the number of c-Fos immunoreactivity (c-Fos-ir) cells in the hippocampus and hypothalamus (Figure 3). One-way ANOVA indicated the effect of APAP overdose on the number of c-Fos-ir cells in both areas analyzed, the DG (*p* < 0.05) and the hypothalamus (*p* < 0.001) (Figure 3).

Post hoc analysis showed an increase in the number of c-Fos-ir cells in the DG of APAPx4-treated mice sacrificed 6 days after the last administration compared with the control and the APAPx4–15 days groups (*^/&^
*p* < 0.05; Figure 3A,F,G).

Regarding the hypothalamus, we observed a higher number of c-Fos-ir cells in the APAPx3, APAPx4–6 h, APAPx4–6 days, and APAPx4–15 days groups than in the control group (*** *p* < 0.001; Figure 3B,I–K). Interestingly, APAPx4-treated mice sacrificed 15 days after the last administration showed a lower c-Fos-ir cell number than that of the other groups treated with repeated APAP administration (APAPx3, APAPx4–6 h, and APAPx4–6 days) (^&/$$/###^
*p* < 0.05/0.01/0.001; Figure 3B,L).

### 2.4. Effect of APAP Overdose on Astroglia in the Hippocampus and Hypothalamus

To investigate the effects of repeated APAP administration on astrogliosis, we evaluated the intensity of GFAP immunoreactivity (GFAP-ir) and the number of astrocytes expressing GFAP in the DG and hypothalamus.

One-way ANOVA indicated the effects of APAP on GFAP-ir cells and GFAP-ir intensity in the DG (*p* < 0.001) (Figure 4A,B).

A greater number of GFAP-ir cells were observed in the DG of APAP-treated mice (APAPx3, APAPx4–6 h, APAPx4–6 days, APAPx4–15 days) compared with the control group (*** *p* < 0.001; Figure 4A,F–I). However, the APAPx4–15 days mice showed a lower number of GFAP-ir cells than the other groups treated with APAP (^$$/&&&/###^ *p* < 0.01/0.001; Figure 4A,I). GFAP-ir intensity was higher in the DG of the APAPx3 and APAPx4–6 days groups than that of the control group (**^/^*** *p* < 0.01/0.001; Figure 4B,F,H). Animals that received four consecutive administrations of APAP and that were sacrificed 15 days later showed lower GFAP-ir intensity than those of the APAPx3 and APAPx4–6 days groups (^#/&&&^ *p* < 0.05/0.001; Figure 4B,I).

Regarding the hypothalamus, one-way ANOVA indicated the effect of repeated APAP administration on GFAP-ir intensity (*p* < 0.001) but not on the number of GFAP-ir cells (Figure 4C,D).

Post hoc analysis showed increased GFAP-ir intensity in the hypothalamus of APAPx3, APAPx4–6 h, and APAPx4–6 days mice compared with the control group (*^/^*** *p* < 0.05/0.001; Figure 4D,K–M). Moreover, the APAPx4–15 days group showed lower GFAP-ir intensity than the APAPx3 and APAPx4–6 days groups (^###/&&&^ *p* < 0.001; Figure 4D,N).

### 2.5. Correlation Analysis between Liver Transaminases and Cell Proliferation and Survival following APAP Overdose

Since elevated liver transaminases are associated with APAP hepatotoxicity [22], we statistically analyzed the correlation between the plasma levels of the liver transaminases gamma-glutamyltransferase (γGT), aspartate aminotransferase (GOT), alanine aminotransferase (GPT), and alkaline phosphatase (ALP); the number of BrdU-ir cells (cell proliferation); and the number of IdU-ir cells (cell survival) after repeated APAP administration (Figure 5).

The results indicated that the plasma levels of γGT, GOT, and GPT, but not ALP, negatively correlated with the number of SGZ BrdU-ir cells (γGT: R = −0.424, F_1,28_ = 6.154, *p* < 0.05; GOT: R = −0.512, F_1,28_ = 9.957, *p* < 0.01; GPT: R = −0.344, F_1,28_ = 3.782, *p* = 0.061). These results suggest that the decrease in SGZ cell proliferation was significantly related to the increase in liver transaminases after repeated APAP administration, an effect that was exacerbated in the APAPx4–6 h group and partially restored in the APAPx4–6 days and APAPx4–15 days groups (Figure 5A,C,E,G). However, cell survival assessed by the number of IdU-ir cells in the hippocampus did not correlate with liver transaminases when the control, APAPx4–6 days, and APAPx4–15 days groups were analyzed (Figure 5B,D,F,H).

The results also indicated that the plasma levels of γGT, GOT, and GPT, but not ALP, negatively correlated with the number of hypothalamic BrdU-ir cells (γGT: R = −0.613, F_1,28_ = 16.90, *p* < 0.001; GOT: R = −0.688, F_1,28_ = 25.27, *p* < 0.0001; GPT: R = −0.463, F_1,28_ = 7.666, *p* < 0.01). These results suggest that the decrease in hypothalamic cell proliferation was significantly related to the increase in liver transaminases induced by repeated APAP administration, an effect that was exacerbated in the APAPx4–6 h group and partially restored in the APAPx4–6 days and APAPx4–15 days groups (Figure 6A,C,E,G). In addition, cell survival was assessed based on the number of IdU-ir cells in the hypothalamus and was correlated with the liver transaminases γGT, GOT, and ALP (γGT: R = −0.752, F_1,16_ = 20.88, *p* < 0.001; GOT: R = −0.444, F_1,16_ = 3.491, *p* = 0.064; ALP: R = −0.651, F_1,16_ = 11.79, *p* < 0.01), but not GPT, when the control, APAPx4–6 days, and APAPx4–15 days groups were analyzed (Figure 6B,D,F,H). These results suggest that the decrease in hypothalamic cell survival was significantly related to the increase in liver transaminases induced by repeated APAP administration, an effect that was exacerbated in the APAPx4–6 days group and partially restored in the APAPx4–15 days group.

### 2.6. Correlation Analysis between Neuronal Activity and Astroglia and Cell Proliferation and Survival following APAP Overdose

We also analyzed the correlation between cell proliferation and survival and neuronal activity and astroglia after repeated APAP administration (Figure 7).

The results indicated that the number of SGZ BrdU-ir cells and the number of hippocampal IdU-ir cells did not correlate with the number of DG c-FOS-ir cells and the number of DG GFAP-ir cells (Figure 7A–D). However, in the hypothalamus, the results also indicated significant correlations between the number of BrdU-ir cells and the number of c-Fos-ir cells (R = −0.712, F_1,28_ = 28.91, *p* < 0.0001) and GFAP-ir cells (R = −0.680, F_1,28_ = 24.08, *p* < 0.0001) (Figure 7E,G). These results suggest that the decrease in hypothalamic cell proliferation was significantly related to an increase in neuronal activity and astrogliosis induced by repeated APAP administration, an effect that was exacerbated in the APAPx4–6 h group and partially restored in the APAPx4–6 days and APAPx4–15 days groups. In addition, significant correlations were also found between the number of IdU-ir cells and the number of c-Fos-ir cells (R = −0.746, F_1,16_ = 20.27, *p* < 0.001) and GFAP-ir cells (R = −0.739, F_1,28_ = 19.30, *p* < 0.001) in the hypothalamus (Figure 7F,H). These results suggest that the decrease in hypothalamic cell survival was significantly related to an increase in neuronal activity and astrogliosis induced by repeated APAP administration, an effect that was exacerbated in the APAPx4–6 days group and partially restored in the APAPx4–15 days group.

### 2.7. Correlation Analysis between Cell Proliferation and Survival following APAP Overdose

We finally analyzed whether changes in cell proliferation in the neurogenic zones are related to changes in cell survival in the hippocampus and hypothalamus after repeated APAP administration. The results indicated that the number of SGZ BrdU-ir cells did not correlate with the number of hippocampal IdU-ir cells (Figure 8A). However, in the hypothalamus, a significant correlation between the number of BrdU-ir cells and the number of IdU-ir cells was found (R = −0.503, F_1,16_ = 5.430, *p* < 0.05) when the control, APAPx4–6 days, and APAPx4–15 days groups were analyzed (Figure 8B).

### 2.8. Effect of APAP Overdose Reduced GSH/GSSG Ratio in the Hypothalamus and Plasma of Control and APAPx3-Treated Mice

Hypothalamus and plasma samples from the control group and the APAPx3 group were analyzed to study the reduced glutathione (GSH)/oxidized glutathione (GSSG) ratio. The mice treated with APAP showed a reduced GSH/GSSG ratio compared with the controls in the hypothalamus (*p* < 0.05), while in plasma, no differences were found (see Appendix A).

## 3. Discussion

Despite the numerous ways in which APAP can affect brain functionality, its effect on adult neurogenesis, a neuroadaptive process strongly susceptible to numerous intrinsic and environmental factors, has not been fully investigated. The main findings of this study are that APAP administered in overdose (750 mg/kg) for consecutive days decreases cell proliferation in the neurogenic areas, the SGZ and the hypothalamus, while also affecting cell survival in the latter area. The reductions in cell proliferation and survival are statistically associated with increases in liver transaminase levels in plasma, as well as astrogliosis (GFAP) and neuronal activity measured based on c-Fos. Interestingly, most of the changes found after the repeated administration of APAP recovered 15 days after the last overdose.

APAP not only has the ability to cross the BBB but it has also been described that the integrity of the BBB is altered by high doses of APAP by disrupting tight junction barrier proteins at the brain microvascular endothelium [24]. In accordance with this, we observed that the expression of occludin, a transmembrane protein of tight junctions that regulates the integrity and permeability of the BBB, leads to a drastic decrease in all groups treated with APAP compared with the control (see Appendix A); furthermore, we did not see a recovery after stopping the treatment in this sense.

Our main hypothesis is that the APAP overdose effect on adult neurogenesis could be due to the direct toxic effect caused by the accumulation of N-acetyl-p-benzoquinone imine (NAPQI) in the brain. Cytochrome P450 2E1 (CYP2E1) is present in brain areas such as the cortex, olfactory bulbs, hippocampus, cerebellum, and brainstem, so brain cells can directly metabolize APAP, producing the toxic reagent NAPQI [3]. To date, there are no studies linking NAPQI toxicity to adult neurogenesis, although there is evidence associating the early consumption of APAP with mid-/long-term neurodevelopment and behavioral defects [25,26,27,28]. Here, we show, for the first time, that there is a negative correlation between adult neurogenesis and the plasma levels of liver damage markers. Thus, we found negative correlations between SGZ cell proliferation and the transaminases γGT and GOT. Likewise, there is a negative correlation between cell proliferation in the hypothalamus and γGT, GOT, and GPT. Furthermore, in the hypothalamus, cell survival also negatively correlates with γGT and ALP. A second finding also indicates that decreased cell proliferation and survival in the hippocampus and hypothalamus induced by APAP overdose are also associated with increased neuronal activity and astrogliosis. Indeed, we found a negative correlation between cell proliferation and survival and the number of cells immunoreactive for c-Fos and GFAP in the hypothalamus. A study by Posadas et al. (2010) [29] showed that rats treated with APAP overdoses had an increased number of apoptotic positive neurons in culture, confirming a direct toxic effect of APAP on brain function that may include neuronal activity and maintenance.

APAP is a safe drug administered in therapeutic doses. However, APAP overdose is the most common cause of acute liver failure (ALF) in many developed countries. NAPQI depletes hepatic GSH, producing mitochondrial dysfunction that ends with the death of hepatocytes. The redox imbalance that occurs within these cells can lead to the excessive production of free radicals capable of crossing the BBB and entering the brain, where they act as neurotoxins. Furthermore, one of the most important consequences of this hepatocellular damage is the cessation of the urea cycle, such that NH_4_^+^ levels in the blood increase radically. This ion is in equilibrium with its non-charged form (NH_3_) in aqueous media such as the bloodstream, so it can cross the BBB and diffuse throughout the CNS. When this molecule reaches the interior of the astrocytes, NH_4_^+^ is used for the synthesis of glutamine [7,30], where high concentrations of this compound lead to cytotoxic effects [8], as we observed in our study with astrogliosis marked by the high expression of GFAP. This fact is one of the many factors that can lead to hepatic encephalopathy (HE) [30,31].

APAP crosses the BBB and is distributed throughout the CNS, where it has analgesic and anti-inflammatory effects [11]. As in the liver, the CYP2E1 enzyme is also expressed in the brain [32] and is found mainly in old areas of the allocortex, such as the hippocampus [33], so this toxic intermediate is produced by brain cells themselves. On the other hand, it has been found that one of the regions of this organ in which GSH levels are most depleted after an overdose of APAP is the hypothalamus [34]. We were able to verify this fact by finding a decrease in the GSH/GSSG ratio found in the hypothalamus of mice exposed to repeated doses of APAP, which also agrees with the existence of an imbalance in GSH homeostasis in neurodegenerative diseases [35]. Therefore, the decrease in cell proliferation and survival that we observed in the hippocampus and hypothalamus, induced by an excess of this drug, is associated with neuronal activity and astrogliosis, and that is why we found a negative correlation between markers of cell proliferation and survival and the number of cells positive for c-Fos and GFAP.

In the brain, APAP inhibits the COX pathway, playing an anti-inflammatory/antipyretic role. However, this process does not occur in the periphery, where the analgesic action of APAP is not related to the inhibition of prostaglandin synthesis [12]. Recently, a COX-independent mechanism of action through cannabinoid/vanilloid signaling was described that explains the analgesic properties of APAP. Upon crossing the BBB, APAP is deacetylated to become p-aminophenol, which, thanks to the fatty acid amide hydrolase (FAAH) enzyme, is conjugated with arachidonic acid to form AM404 [36,37].

AM404 is a ligand of cannabinoid receptor 1 (CB1) and an inhibitor of AEA uptake into cells, thereby increasing cannabinoid tone [3,36,38]. Recent studies have implicated the endocannabinoid system in neurogenic processes. Thus, endocannabinoid signaling controls neuronal proliferation and survival as a neuroprotective response to brain insults [39,40,41,42]. Several studies have shown that the activation of CB1 receptors facilitates cell proliferation and survival in the hippocampus and hypothalamus; these studies have also shown the direct role of the CB1 receptor in neurogenesis since the number of BrdU-labeled cells can be significantly reduced in the hippocampal SGZ and SVZ of CB1 knockout mice [39,42,43,44]. Thus, we expect that AM404, an AEA reuptake inhibitor that enhances the effects of AEA in vivo through the cannabinoid CB1 receptor, should increase cell proliferation and survival after the administration of APAP. However, this hypothesis is not consistent with our results, which demonstrate that the excessive and continued consumption of APAP reduces adult neurogenesis (cell proliferation and survival) in the SGZ and hypothalamus. Several factors must be considered to explain our results. First, the inhibitory effect of APAP overdose on adult neurogenesis may be a consequence of CB1 desensitization in the context of elevated AEA tone. Second, there is a possibility of APAP interactions with independent CB1 pathways, including vanilloid, serotonergic, and nitric oxide systems [37,45,46]. Indeed, both anandamide and AM404 are agonists of the capsaicin receptor TRPV1, a vanilloid receptor that mediates decreased cell proliferation [43,47]. 

Given all these premises, we conclude that there is a direct toxic effect of excess paracetamol on the brain, decreasing adult neurogenesis in the hippocampus and hypothalamus and increasing neuroinflammation. These effects do not seem to be mediated by metabolites generated in the transformation of APAP as it crosses the BBB and, therefore, would not involve the action of the cannabinoid and vanilloid signaling systems. To confirm the hypothesis based on our results, future studies using antioxidants that prevent the accumulation of NAPQI are necessary.

The alterations in cell proliferation and survival caused by drugs in different neurogenic niches can be transient and, therefore, can be restored after the cessation of their administration, as occurs, for example, in studies of cocaine addiction. This does not mean that there are no long-term neuroadaptations induced by the drug (genes related to neuroplasticity, brain-derived neurotrophic factor (BDNF), cellular morphology, etc.) or changes at a cognitive level [48]. In future studies, it would be interesting to analyze these possible long-term neuroadaptations induced by paracetamol overdose, performing, for example, behavioral tests during treatment with APAP and after its cessation.

From a clinical point of view, drug-induced hepatotoxicity is an important disease, as APAP is the most frequent cause in relation to ingested doses [49]. Generally, APAP is consumed alone or in combination with other medication, frequently in self-medication without medical advice, thus causing the majority of cases of APAP poisoning.

The liver toxicity of APAP has been widely studied, but the toxic effect of this drug on the brain has been less investigated. More studies are needed to clarify the direct damage of APAP in the brain (by crossing the BBB) and the indirect damage associated with worsening liver function. Also, deeper knowledge of the pathways that mediate APAP toxicity in the brain is necessary.

Our results provide information in this regard, helping to comprehensively understand APAP’s mechanism of toxic action in the brain, which may be fundamental in the development of new therapeutic approaches.

## 4. Materials and Methods

### 4.1. Ethics Statement

The experimental procedures with animals were carried out following the recommendations of European Communities Directive 2010/63/EU and Spanish legislation (Real Decreto 53/2013, BOE 34/11370-11421, 2013) regulating the care and use of laboratory animals. The protocol was approved by the Animal Ethics Committee of the University of Málaga (Ref. no. 24-2015-A). All efforts were made to minimize animal suffering and to reduce the number of animals used. Animal studies comply with the ARRIVE guidelines.

### 4.2. Animal Model

Male Crl:CD1 (ICR) mice (approximately 25–30 g, 3–4 months old) were purchased from Charles Rivers Laboratories (Barcelona, Spain). They were housed in cages maintained in standard conditions (Centro de Experimentación y Conducta Animal, Universidad de Málaga) at 20 ± 2 °C room temperature, 40 ± 5% relative humidity, and a 12h light⁄dark cycle with a dawn/dusk effect. Water and standard rodent chow (Prolab, Saint Louis, MO, USA; RMH 2500, 2.9 kcal/g) were available ad libitum.

The animals were handled daily for 10 min and habituated to an oral gavage procedure for 1 week before experimentation in order to minimize stress effects.

### 4.3. Acetaminophen Treatment

Acetaminophen (APAP, cat. no. A7085, Sigma-Aldrich, Saint Louis, MO, USA) was dissolved in a vehicle containing 0.5% dimethyl sulfoxide (DMSO) in sterile 0.9% NaCl solution just before each experiment. The APAP was orally gavage-administrated in a final concentration of 750 mg/kg body weight.

Mice received repeated administration of APAP and were sacrificed 6 h or several days after the last administration to study the recovery. Mice were randomly divided into five groups (N = 6): (1) repeated vehicle administration for 4 days (control group); (2) repeated APAP administration (750 mg/kg/day) for 3 days and sacrificed 6 h after the last administration (APAPx3); (3) repeated APAP administration (750 mg/kg/day) for 4 days and sacrificed 6 h after the last administration (APAPx4–6 h); (4) repeated APAP administration (750 mg/kg/day) for 4 days and sacrificed 6 days after the last administration (APAPx4–6 days); and (5) repeated APAP administration (750 mg/kg/day) for 4 days and sacrificed 15 days after the last administration (APAPx4–15 days). Figure 9 shows the administration scheme and the experimental groups used in this study.

### 4.4. IdU and BrdU Administration

To analyze cell proliferation and survival, 5′-bromo-2′-deoxyuridine (BrdU, cat. no. B5002, Sigma-Aldrich, Sant Louis, USA) and 5′-iodo-2′-deoxyuridine (IdU, cat. no. I7125, Sigma-Aldrich) were administered intraperitoneally (i.p.) in a sterile 0.9% NaCl solution at a dose of 50 mg/kg/day and 42.75 mg/kg/day, respectively.

BrdU was administered to all experimental groups. BrdU was injected i.p. twice per day at 10 h intervals for 3 consecutive days before sacrifice. Thus, BrdU was used as a cell proliferation marker. IdU was injected i.p. twice per day at 10 h intervals for 3 consecutive days on days 2–4 of the experiment. Using this regimen of administration, IdU was administered to the control, APAPx4–6 days, and APAPx4–15 days groups only. Thus, IdU was used as a cell survival marker (Figure 9).

### 4.5. Sample Collection

Animals were fasted for 12 h before sacrifice. Previous to sacrifice, all animals were anesthetized (sodium pentobarbital, 50 mg/kg body weight, i.p.) in a room separate from the other experimental animals. The blood samples were transcardially collected into tubes containing heparin and centrifuged (1600× *g* for 10 min, 4 °C), and the plasma was stored at −80 °C for biochemical analysis. Another batch of animals was transcardially perfused with 4% formaldehyde in 0.1 M phosphate buffer (PB). Brains were dissected out and kept in the same fixative solution overnight at 4 °C.

The brains were then cut into 30 μm thick coronal sections by using a sliding microtome (Leica, Wetzlar, Germany; VT1000S) and divided into 6 parallel series. Sections were stored at 4 °C in PB with 0.002% (*w*/*v*) sodium azide until they were used for immunostaining.

### 4.6. Biochemical Analysis

ALP and the hepatic enzymes ALT or GPT, AST or GOT, and γGT were analyzed using commercial kits according to the manufacturer’s instructions in a Hitachi 737 Automatic Analyzer (Hitachi Ltd., Tokyo, Japan). In all cases, a calibration curve and internal controls were included in each assay.

### 4.7. Immunohistochemistry and Immunofluorescence

Free-floating coronal sections from −2.16 to −4.20 Bregma levels (hippocampus and hypothalamus) were selected from one of the six parallel series obtained from each brain from the five experimental groups [50].

Floating brain sections were incubated overnight at 4 °C with rat anti-BrdU (1:2000; Accurate Chemical & Scientific, Nueva York, NY, USA; OBT0030 F), mouse anti-IdU (1:500; Sigma-Aldrich, Sant Louis, MI, USA), rabbit anti-c-Fos antibody (1:10,000 Calbiochem, San Diego, CA, USA; PC38), mouse anti-glial fibrillaric acidic protein (GFAP) (1:500, Sigma, Sant Louis, MI, USA; cat. no. G3893), and mouse anti-occludin (1:200; Invitrogen, Waltham, MA, USA; OC-3F10) antibody [42].

The following day, the sections were incubated in their respective secondary antibodies for 90 min: biotinylated donkey anti-rat IgG (1:500, Novex by Life Technologies, Carlsbad, CA, USA, cat. No. A18743), biotinylated goat anti-mouse IgG (1:500, Sigma, Sant Louis, MI, USA; cat. No. B7264), or biotinylated donkey anti-rabbit IgG (1:500, Amersham, Little Chalfont, UK; cat. no. RPN 1004). The sections were then incubated in ExtrAvidin peroxidase (Sigma) diluted 1:2000 in darkness at room temperature for 1 h. Finally, immunolabeling was revealed with 0.05% diaminobenzidine (DAB; Sigma), 0.05% nickel ammonium sulfate, and 0.03% H_2_O_2_ in PBS.

For immunofluorescence, sections were incubated for 2 h at room temperature with secondary antibody goat anti-mouse IgG labeled with Alexa 488 (1:200; Invitrogen, Waltham, MA, USA; A-11029).

### 4.8. Cell Counting

BrdU, IdU, c-Fos, and GFAP-ir nuclei and cells that came into focus were manually counted from Bregma −2.16 to −4.20 mm at hippocampal and hypothalamic levels (Paxinos & Watson, 2007) using a standard optical microscope with a 40× objective (Nikon Instruments Europe B.V., Amstelveen, The Netherlands) coupled to the NIS-Elements Imaging Software 3.00 (Nikon, Tokyo, Japan).

Focusing on the hippocampus, BrdU-ir nuclei were counted in the SGZ of the DG, while IdU-ir nuclei and c-Fos and GFAP-ir cells were counted in the whole hippocampus (DG, CA3, and CA1 areas). Focusing on the hypothalamus, counting was performed in the paraventricular (PVH), ventromedial (VMH), and arcuate (ARC) nuclei and median eminence.

Immunostained cells located in the uppermost side that came into focus while moving down through the thickness of the section were counted. Overall, quantification was expressed as the average number of cells per area (mm^2^) in each experimental group.

### 4.9. Quantification of Immunoreactivity

Densitometric analysis of the GFAP immunoreactivity was conducted on the three hippocampal areas (DG, CA3, and CA1) and hypothalamus from Bregma −2.16 to −4.20 mm [50]. High-resolution digital microphotographs of representative areas were taken with a 10× objective under the same conditions of light and brightness/contrast with an Olympus BX41 microscope equipped with an Olympus DP70 digital camera (Tokyo, Japan). Quantification was determined using the analysis software ImageJ 1.38X (NIH, USA).

### 4.10. Determination of Reduced and Oxidized Glutathione

Quantification of the reduced (GSH), free GSH, and oxidized GSH (GSSG) in the hypothalamus and liver were measured using a commercial kit according to the manufacturer’s instructions (Invitrogen, Waltham, MA, USA; Cat# EIAGSHC).

### 4.11. Statistical Analysis

The GraphPad Prism 9 software was used. All data are expressed as means ± standard error of the mean (SEM) (N = 6 per experimental group). Animal model data were analyzed by one-way ANOVA. Subsequent multiple comparisons between groups were carried out using Tukey adjustments. To analyze whether repeated APAP administration induced changes in the association between parameters, Pearson’s correlation analysis was performed, and *Rho* values (goodness-of-fit) were calculated. A *p*-value less than 0.05 indicates statistical significance.

## Figures and Tables

**Figure 1 ijms-25-01964-f001:**
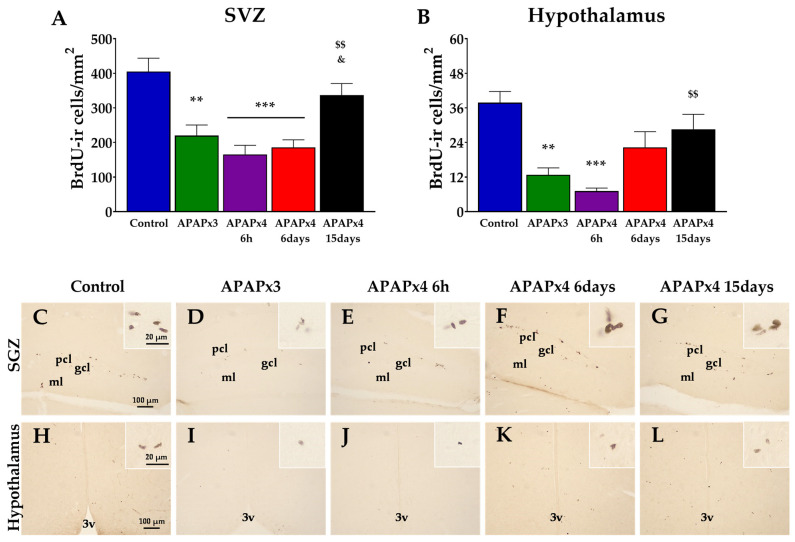
Effect of repeated administration of acetaminophen (APAP) at a dose of 750 mg/kg/day on cell proliferation analyzed by 5′-bromo-2′-deoxyuridine immunohistochemistry (BrdU-ir) in the subgranular zone (SGZ) of the dentate gyrus (**A**) and the hypothalamus (**B**) of male mice. The histogram represents the means ± standard error of the mean (SEM) per section of BrdU-ir nuclei (N = 6 samples per group). Representative micrographs show magnification views of the typical clustering of newborn cells at the inner border of the SGZ (**C**–**G**) and hypothalamus (**H**–**L**). 3v, third ventricle; gcl, granular cell layer; ml, molecular layer; pcl, polymorphic cell layer. One-way ANOVA: (**) *p* < 0.01, and (***) *p* < 0.001 vs. control group. (^$$^) *p* < 0.01 vs. APAPx4–6 h group. (^&^) *p* < 0.05 vs. APAPx4–6days group.

**Figure 2 ijms-25-01964-f002:**
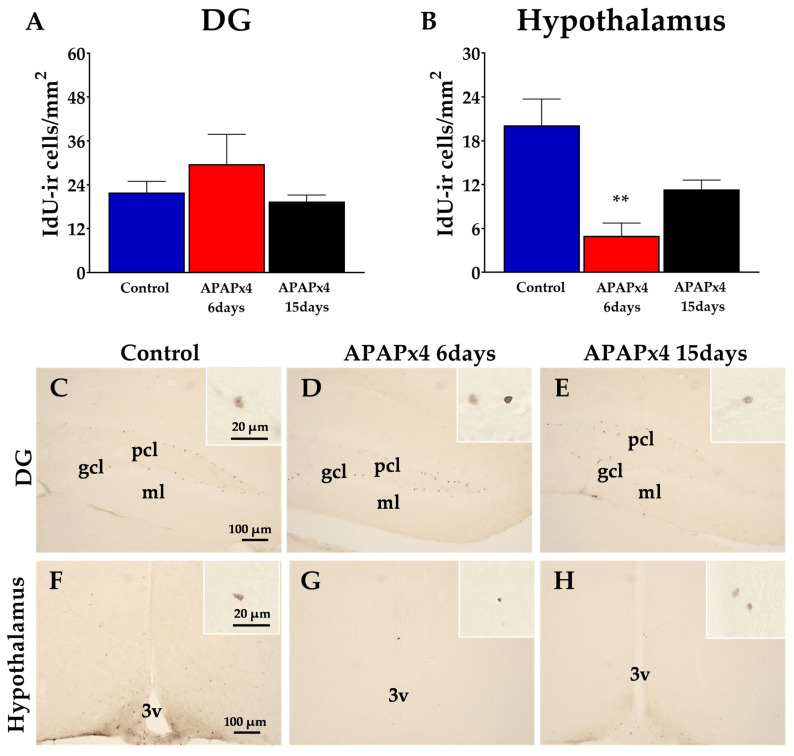
Effect of repeated administration of acetaminophen (APAP) at a dose of 750 mg/kg/day on cell survival analyzed by 5′-iodo-2′-deoxyuridine immunohistochemistry (IdU-ir) in the dentate gyrus (DG) of the hippocampus (**A**) and the hypothalamus (**B**) in male mice. The histogram represents the means ± standard error of the mean (SEM) per section of IdU-ir nuclei (N = 6 samples per group). Representative micrographs show magnification views of the typical clustering of survival cells at the inner border of the subgranular zone (SGZ) of the DG (**C**–**E**) and the hypothalamus (**F**–**H**). 3v, third ventricle; gcl, granular cell layer; ml, molecular layer; pcl, polymorphic cell layer. One-way ANOVA: (**) *p* < 0.01 vs. control group.

**Figure 3 ijms-25-01964-f003:**
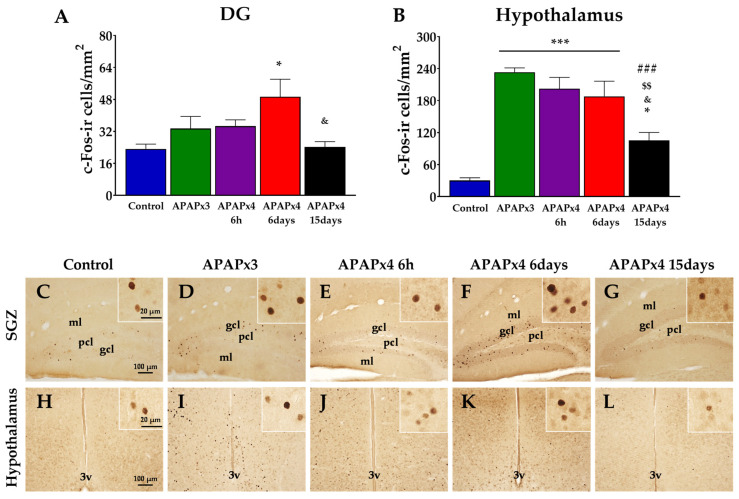
Effect of repeated administration of acetaminophen (APAP) at a dose of 750 mg/kg/day on neuronal activity analyzed by c-Fos immunohistochemistry in the dentate gyrus (DG) of the hippocampus (**A**) and the hypothalamus (**B**) of male mice. The histogram represents the means ± standard error of the mean (SEM) per section of c-Fos immunoreactivity (c-Fos-ir) cells (N = 6 samples per group). Representative micrographs show magnification views of c-Fos-ir cells in the hippocampus (**C**–**G**) and the hypothalamus (**H**–**L**). 3v, third ventricle; gcl, granular cell layer; ml, molecular layer; pcl, polymorphic cell layer. One-way ANOVA: (*) *p* < 0.05 and (***) *p* < 0.001 vs. control group. (^###^) *p* < 0.001 vs. APAPx3 group. (^$$^) *p* < 0.01 vs. APAPx4–6 h group. (^&^) *p* < 0.05 vs. APAPx4–6 days group.

**Figure 4 ijms-25-01964-f004:**
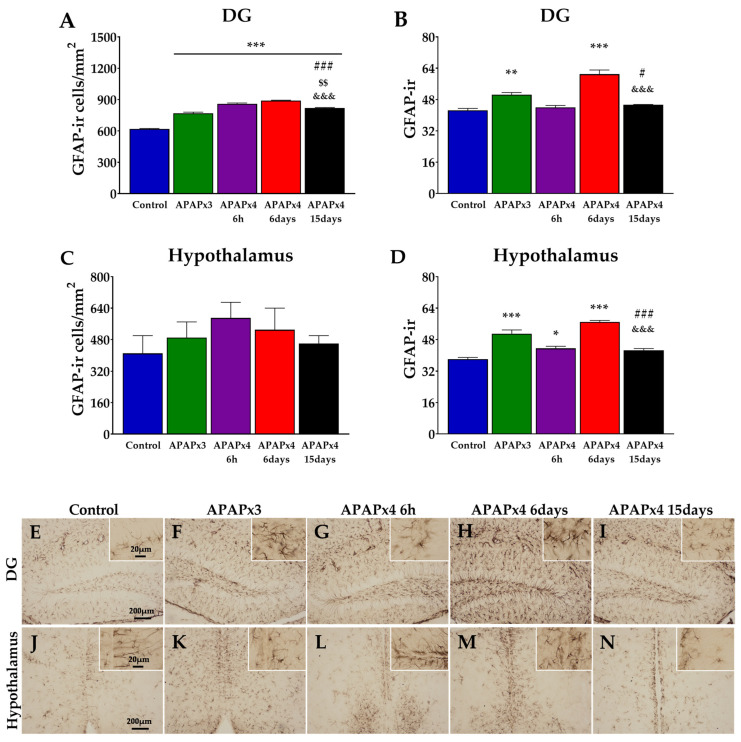
Effect of repeated administration of acetaminophen (APAP) at a dose of 750 mg/kg/day on astrogliosis analyzed by glial fibrillary acidic protein (GFAP) immunohistochemistry in the dentate gyrus (DG) of the hippocampus (**A**,**B**) and the hypothalamus (**C**,**D**) of male mice. The histogram represents the means ± standard error of the mean (SEM) per section of GFAP immunoreactivity (GFAP-ir) cells and GFAP-ir intensity (n = 6 samples per group). Representative micrographs show magnification views of GFAP-ir cells and GFAP-ir intensity in the DG (**E**–**I**) and hypothalamus (**J**–**N**). One-way ANOVA: (*) *p* < 0.05, (**) *p* < 0.01, and (***) *p* < 0.001 vs. control group. (^#^) *p* < 0.05 and (^###^) *p* < 0.001 vs. APAPx3 group. (^$$^) *p* < 0.01 vs. APAPx4–6 h group. (^&&&^) *p* < 0.001 vs. APAPx4–6 days group.

**Figure 5 ijms-25-01964-f005:**
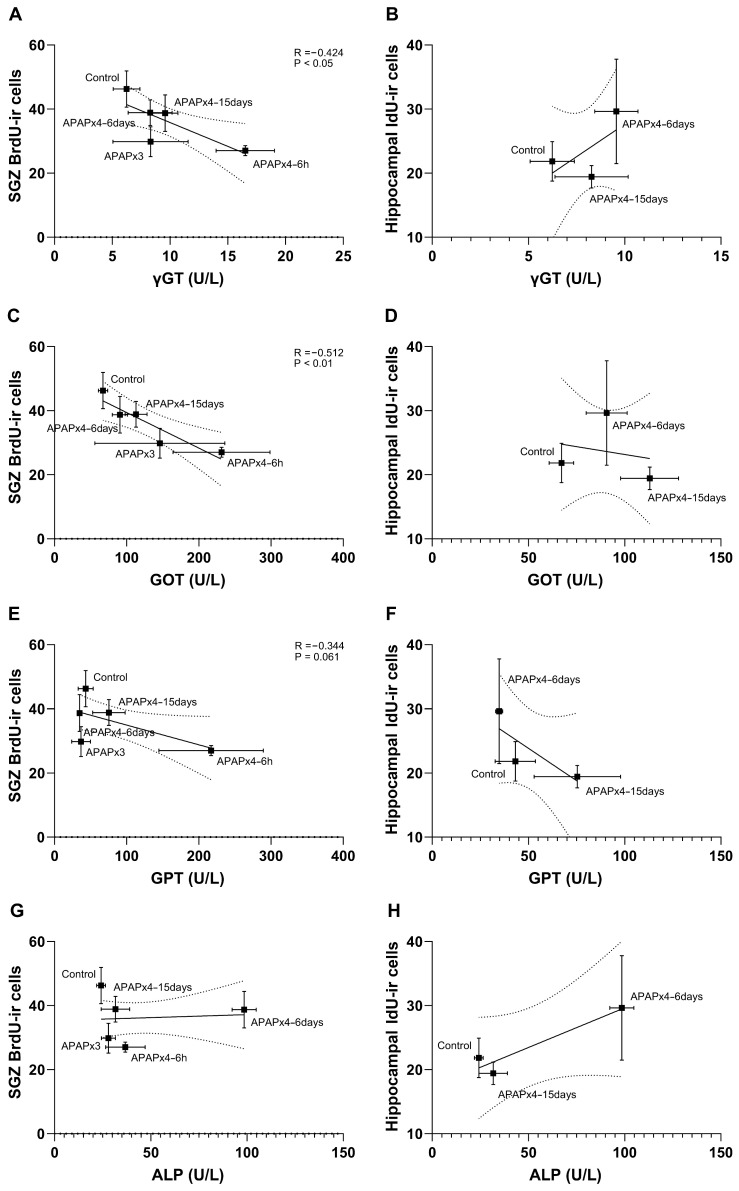
Correlation analysis between plasma levels of liver transaminases gamma-glutamyltransferase (γGT; **A**,**B**), aspartate aminotransferase (GOT; **C**,**D**), alanine aminotransferase (GPT; **E**,**F**), and alkaline phosphatase (ALP; **G**,**H**) and cell proliferation in the subgranular zone (SGZ), assessed by the number of 5′-bromo-2′-deoxyuridine immunoreactivity (BrdU-ir) cells, and cell survival in the hippocampus, assessed by the number 5′-bromo-2′-deoxyuridine immunoreactivity (IdU-ir) cells, in mice that received vehicle (control group) or acetaminophen (APAP) at a dose of 750 mg/kg for 3 and 4 consecutive days and that were sacrificed 6 h (APAPx3 and APAPx4–6 h groups), 6 days (APAPx4–6 days group), and 15 days (APAPx4–15 days group) after the last administration. The scatter (XY) plots represent the means ± standard error of the mean (SEM) (N = 6 per experimental group). Plotted (solid) lines between mean points represent the correlative changes between the two variables represented. Dashed lines represent error of the plotted lines. *Rho* values (goodness-of-fit) were calculated. A *p*-value less than 0.05 indicates the statistical significance of the correlation.

**Figure 6 ijms-25-01964-f006:**
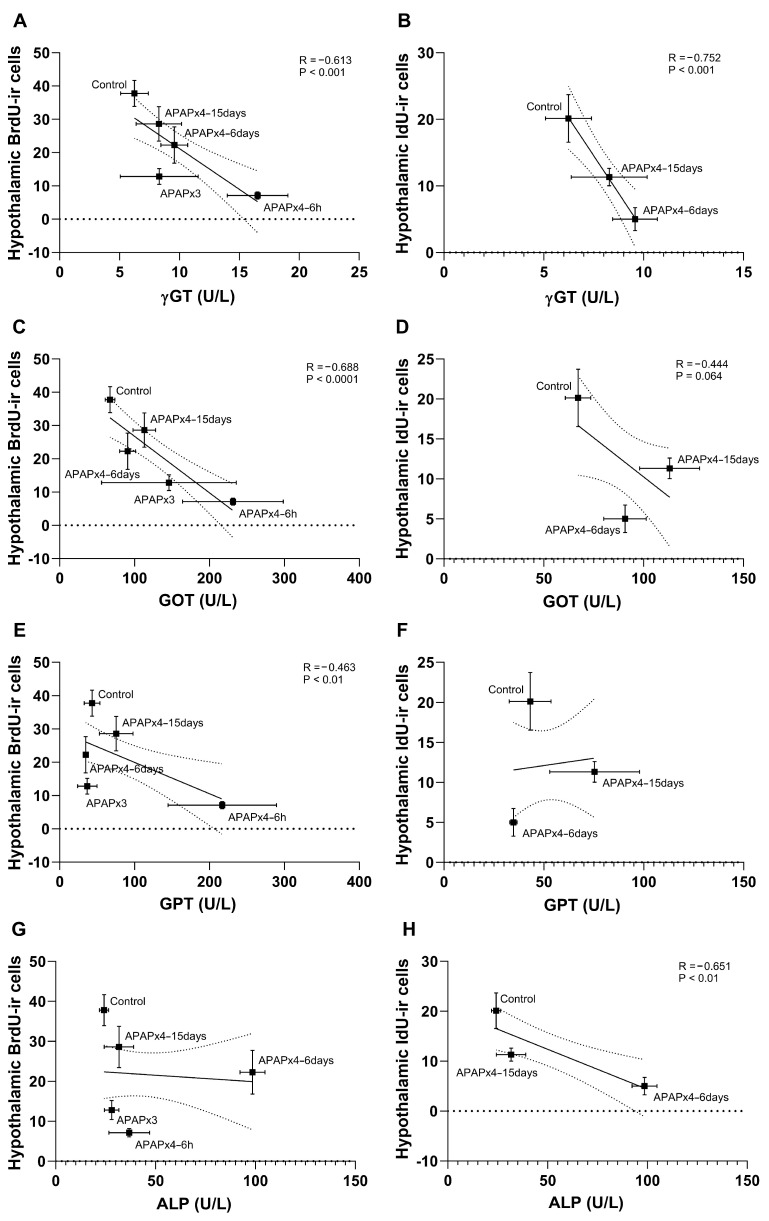
Correlation analysis between plasma levels of the liver transaminases gamma-glutamyltransferase (γGT; **A**,**B**), aspartate aminotransferase (GOT; **C**,**D**), alanine aminotransferase (GPT; **E**,**F**), and alkaline phosphatase (ALP; **G**,**H**) and cell proliferation, assessed by the number of 5′-bromo-2′-deoxyuridine immunoreactivity (BrdU-ir) cells, and cell survival, assessed by the number of 5′-iodo-2′-deoxyuridine immunoreactivity (IdU-ir) cells, in the hypothalamus of mice that received saline (control group) or acetaminophen (APAP) at a dose of 750 mg/kg for 3 and 4 consecutive days and that were sacrificed 6 h (APAPx3 and APAPx4–6 h groups), 6 days (APAPx4–6 days group), and 15 days (APAPx4–15 days group) after the last administration. The scatter (XY) plots represent the means ± standard error of the mean (SEM) (N = 6 per experimental group). Plotted (solid) lines between mean points represent the correlative changes between the two variables represented. Dashed lines represent error of the plotted lines. *Rho* values (goodness-of-fit) were calculated. A *p*-value less than 0.05 indicates the statistical significance of the correlation.

**Figure 7 ijms-25-01964-f007:**
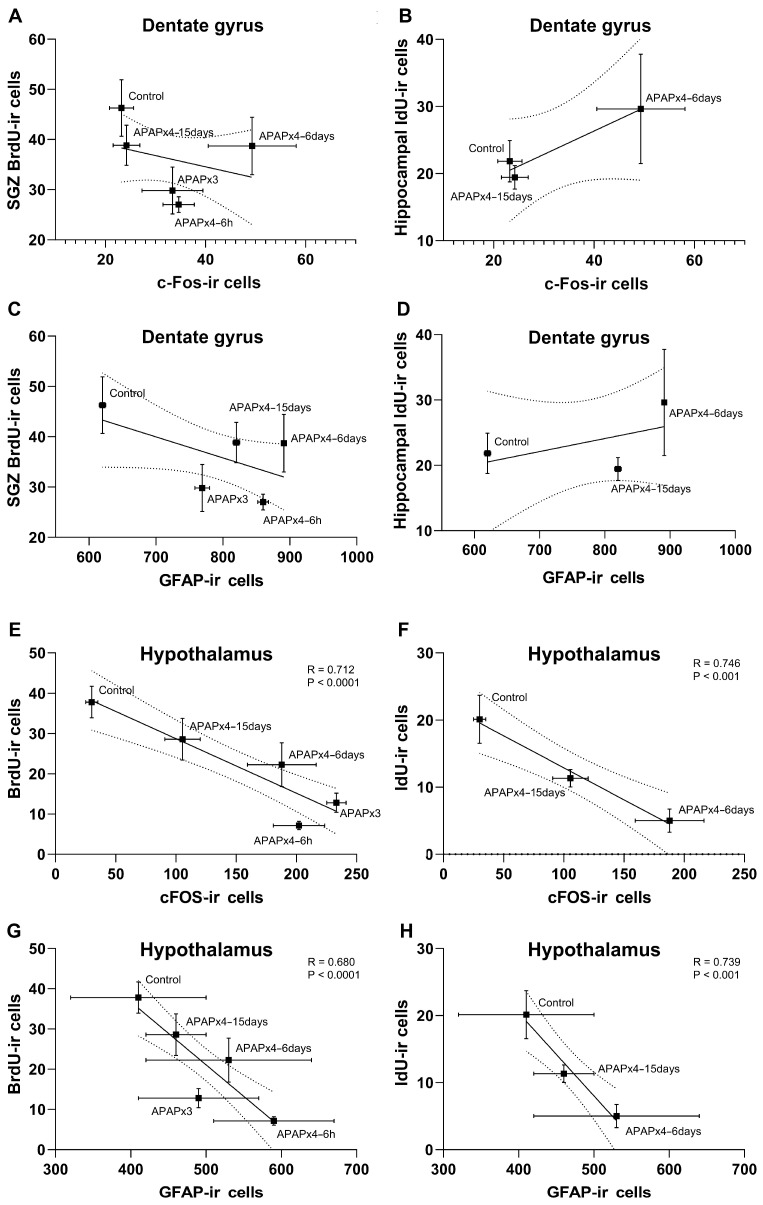
Correlation analysis between cell proliferation, assessed by the number of 5′-bromo-2′-deoxyuridine immunoreactivity (BrdU-ir) cells; cell survival, assessed by the number 5′-iodo-2′-deoxyuridine immunoreactivity (IdU-ir) cells; neuronal activity, assessed by the number of c-Fos immunoreactivity (c-Fos-ir) cells; and astroglia, assessed by the number of glial fibrillary acidic protein immunoreactivity (GFAP-ir) cells, in the hippocampus (**A**–**D**) and hypothalamus (**E**–**H**) of mice that received saline (control group) or acetaminophen (APAP) at a dose of 750 mg/kg for 3 and 4 consecutive days and that were sacrificed 6 h (APAPx3 and APAPx4–6 h groups), 6 days (APAPx4–6 days group), and 15 days (APAPx4–15 days group) after the last administration. The scatter (XY) plots represent the means ± standard error of the mean (SEM) (N = 6 per experimental group). Plotted (solid) lines between mean points represent the correlative changes between the two variables represented. Dashed lines represent error of the plotted lines. *Rho* values (goodness-of-fit) were calculated. A *p*-value less than 0.05 indicates the statistical significance of the correlation.

**Figure 8 ijms-25-01964-f008:**
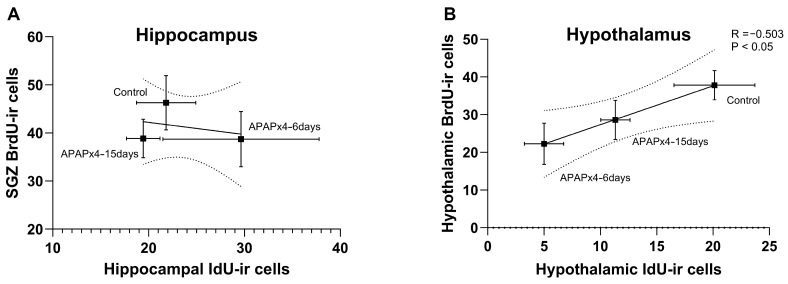
Correlation analysis between cell proliferation, assessed by the number of 5′-bromo-2′-deoxyuridine immunoreactivity (BrdU-ir) cells, and cell survival, assessed by the number 5′-bromo-2′-deoxyuridine immunoreactivity (IdU-ir) cells, in the hippocampus (**A**) and hypothalamus (**B**) of mice that received saline (control group) or acetaminophen (APAP) at a dose of 750 mg/kg for 4 consecutive days and that were sacrificed 6 days (APAPx4–6 days group) and 15 days (APAPx4–15 days group) after the last administration. The scatter (XY) plots represent the means ± standard error of the mean (SEM) (N = 6 per experimental group). Plotted (solid) lines between mean points represent the correlative changes between the two variables represented. Dashed lines represent error of the plotted lines. *Rho* values (goodness-of-fit) were calculated. A *p*-value less than 0.05 indicates the statistical significance of the correlation.

**Figure 9 ijms-25-01964-f009:**
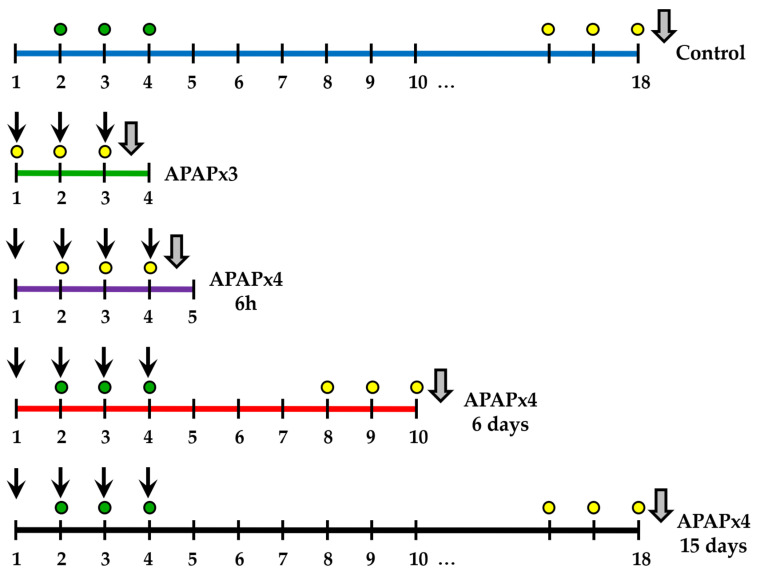
Schematic timeline and experimental groups of repeated administration of acetaminophen (APAP) at a dose of 750 mg/kg/day for 3 (green line) or 4 (purple line) consecutive days and cessation of APAP administration for 6 (red line) or 15 (black line) consecutive days. Black arrow: APAP administration; gray arrow: sacrifice; yellow circle: 5′-bromo-2′-deoxyuridine immunoreactivity (BrdU) administration (50 mg/kg/day); green circle: 5′-iodo-2′-deoxyuridine immunoreactivity IdU administration (42.75 mg/kg/day).

## Data Availability

The data that support the findings of this study are available from the corresponding authors upon reasonable request.

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
