# Peer review of "Inhibition of Adult Neurogenesis in Male Mice after Repeated Exposure to Paracetamol Overdose"

_ijms, 2024, doi:10.3390/ijms25041964_

Round 1
Reviewer 1 Report
Comments and Suggestions for Authors
The Article titled "Inhibition of adult neurogenesis in male mice after repeated exposure to paracetamol overdose" is an interesting work, but major information missed to conclude.
The article is well-organized and presents promising results.
The author has used an adequate assessment that relatively responds to the objectives of the article.
However :
The author has to enrich the introduction with a more scientific explanation and relationship between liver and brain regulation.
The author based the discussion on GSH depletion and inflammation, but it will be essential to evaluate GSH in the liver and brain and also evaluate the inflammation markers in the plasma to have a clear idea and conclusion.
Additionally, it will be essential to evaluate the RNA expression of the key genes in this process.
The author also based the discussion on the NH4+ and urea cycle. So, evaluating this level in plasma will be important to empower the conclusion.
The author needs to include the signaling pathway regulation to explain more the effects in the discussion.
It is also crucial to add a scientific explanation of the role of time in the return to normal.
The author must show the major revisions made in the text by highlighting the changes in a different colored text.
It is imperative to consider all these remarks to reinforce the manuscript's quality and conclude more accurately.
Reviewer 2 Report
Comments and Suggestions for Authors
The study analyzed the toxic effects of acetaminophen (N-acetyl-para-aminophenol, APAP) overdose (750 mg/kg/day), for 3 and 4 consecutive days and after cessation of APAP administration for 6 and 15 days in a mouse model, on cell proliferation and survival in two relevant neurogenic zones: the subgranular zone of the dentate gyrus and the hypothalamus. The involvement of liver damages, neuronal activities and astroglia were also evaluated. In general, the experiments were well-performed and the manuscript was well-written. It may be published pending some minor revisions.
1) All abbreviations should be introduced at the first time of appearance, such as "GFAP" in Abstract, "pcl", "gcl", "ml" and "3v" in the legends of Figures 1-4, and "γGT, GOT, GPT and ALP " in the legend of Figure 5.
2) In Abstract, the authors should clearly state that the experiments were performed in a mouse model.
3) Clinical implications for human APAP should be discussed.
Additional comments:
• Do you consider the topic original or relevant in the field? Does it address a specific gap in the field?
Yes, it is original and relevant.
• What does it add to the subject area compared with other published material?
APAP is an analgesic and antipyretic drug, which overdose would cause serious liver damage. APAP can cross the blood-brain barrier and affects brain function in numerous ways. However, its effects on adult neurogenesis have not been investigated before.
• What specific improvements should the authors consider regarding the methodology? What further controls should be considered?
The controls were fine. No further comments.
Comments on the Quality of English LanguageLine 28: "...was also evaluated." --> were also evaluated.
In the legends of Figures 5-8, I cannot understand the sentence "Plotted lines between mean points represent the correlative 207 changes between the two variables represented".
Reviewer 3 Report
Comments and Suggestions for Authors
Dear Authors,
The manuscript titled “Inhibition of adult neurogenesis in male mice after repeated exposure to paracetamol overdose” is very interesting, pointing out interesting results on paracetamol effect in male mice. Moreover, the manuscript is well written and logically constructed. However, I have some questions and suggestions in order to better improve the quality of the paper.
1- The Authors, correctly, not only focused their research on direct neurodegeneration but also showing possibile effect on glia compartment. Concerning figure 4, how the Author might explain that GFAP immunoreation in APAPx4 6h treatment is less than APAPx3 experimental point?
2- Furthermore, did the Authors analysed the microglial immunoreaction? It should be interesting to know if microglial cells are more sensitive to paracetamol and, secondly, if the gliosis restoration occurred at the same time or not.
3- Still concerning the GFAP immunoreaction, did the Authors analysed the increasing of cells number? The time and dose-dependent GFAP increment was only due by the cell morphology?
4- The Authors indicated that BBB can be crossed by APAP (reference n. 6). However, the reference quoted hypotheses that since APAP is lipophilic can easily cross the BBB. I suggest the Authors to quote more relevant paper on this topic (such as: doi: 10.1542/peds.2006-3378; doi: 10.1016/j.phrs.2016.02.020). Furthermore, it should be very interesting if the Authors can show BBB dysregulation in the mice’s brain after APAP administration, and more interesting if the BBB back to control level stopping the APAP treatment. Immunolabelling with antibody against tight junctions proteins, such as claudins, occludin or zonula occludens are suggested. The Authors should consider that the altered BBB induce a not controlled entrance of any substances in the brain parenchyma thus leading to deleterious chronic and indirect side effect.
5- I suggest immunofluorescent staining for GFAP immuoreaction in order to better visualize differences. Furthermore, elargment insets or at least 20x images instead 10x would be preferred in order to better visualize glial alterations.
Typos:
Figure 2, panel E: APAPX4 15days, should be written as APAPx4.
Round 2
Reviewer 1 Report
Comments and Suggestions for Authors
In this version of the Article titled "Inhibition of adult neurogenesis in male mice after repeated exposure to paracetamol overdose» the author has relatively responded to remarks and criticisms, especially in the scientific explanation.
Unfortunately, the author did not make much effort in carrying out the requested measures (for technical and economic reasons) because it would have been beneficial to reinforce the manuscript's quality and conclude more accurately.
in any case, it is a good work with good results which I recommend its publication in the international journal of molecular sciences.